# Classification with Valid and Adaptive Coverage

**Yaniv Romano**[*]
Department of Statistics
Stanford University
Stanford, CA, USA
yromano@stanford.edu

**Matteo Sesia**[*]
Department of Data Sciences and Operations
University of Southern California
Los Angeles, CA, USA
sesia@marshall.usc.edu

**Emmanuel J. Candès**
Departments of Mathematics and of Statistics
Stanford University
Stanford, CA, USA
candes@stanford.edu

## Abstract

Conformal inference, cross-validation+, and the jackknife+ are hold-out methods that can be combined with virtually any machine learning algorithm to construct prediction sets with guaranteed marginal coverage. In this paper, we develop specialized versions of these techniques for categorical and unordered response labels that, in addition to providing marginal coverage, are also fully adaptive to complex data distributions, in the sense that they perform favorably in terms of approximate conditional coverage compared to alternative methods. The heart of our contribution is a novel conformity score, which we explicitly demonstrate to be powerful and intuitive for classification problems, but whose underlying principle is potentially far more general. Experiments on synthetic and real data demonstrate the practical value of our theoretical guarantees, as well as the statistical advantages of the proposed methods over the existing alternatives.

## 1 Introduction

Imagine we have $n$ data samples $\{(X_i, Y_i)\}_{i=1}^n$ with features $X_i \in \mathbb{R}^p$ and a discrete label $Y_i \in \mathcal{Y} = \{1, 2, \ldots, C\}$. The samples are drawn exchangeably (e.g., i.i.d., although exchangeability alone is sufficient) from some unknown distribution $P_{XY}$. Given such data and a desired coverage level $1 - \alpha \in (0, 1)$, we seek to construct a prediction set $\hat{\mathcal{C}}_{n,\alpha} \subseteq \mathcal{Y}$ for the *unseen* label of a new data point $(X_{n+1}, Y_{n+1})$, also drawn exchangeably from $P_{XY}$, achieving marginal coverage; that is, obeying

$$\mathbb{P}\left[Y_{n+1} \in \hat{\mathcal{C}}_{n,\alpha}(X_{n+1})\right] \geq 1 - \alpha. \tag{1}$$

The probability above is taken over all $n + 1$ data points, and we ask that (1) holds for any fixed $\alpha$, $n$, and $P_{XY}$. While marginal coverage has the advantage of being both desirable and practically achievable, it unfortunately does not imply the stronger notion of conditional coverage:

$$\mathbb{P}\left[Y_{n+1} \in \hat{\mathcal{C}}_{n,\alpha}(x) \mid X_{n+1} = x\right] \geq 1 - \alpha. \tag{2}$$

The latter asks for valid coverage conditional on a specific observed value of the features $X$. It is already known that conditional coverage cannot be achieved in theory without strong modeling

---

[*]Equal contribution.

assumptions [1, 23], which we are not willing to make in this paper. That said, it is undeniable that conditional coverage would be preferable. We thus seek to develop classification methods that are provably valid in the marginal sense (1) and also attempt to sensibly approximate conditional coverage (16). At the same time, we want powerful predictions, in the sense that the cardinality of $\hat{C}$ should be as small as possible.

## 1.1 The oracle classifier

Imagine we have an *oracle* with perfect knowledge of the conditional distribution $P_{Y|X}$ of $Y$ given $X$. This would of course give the problem away; to be sure, we would define optimal prediction sets $C_\alpha^{\text{oracle}}(X_{n+1})$ with conditional coverage as follows: for any $x \in \mathbb{R}^p$, set $\pi_y(x) = \mathbb{P}[Y = y \mid X = x]$ for each $y \in \mathcal{Y}$. Denote by $\pi_{(1)}(x) \geq \pi_{(2)}(x) \geq \ldots \geq \pi_{(C)}(x)$ the order statistics for $\pi_y(x)$. For simplicity, let us assume for now that there are no ties; we will relax this assumption shortly. For any $\tau \in [0, 1]$, define the *generalized conditional quantile* function[2]

$$L(x; \pi, \tau) = \min\{c \in \{1, \ldots, C\} \; : \; \pi_{(1)}(x) + \pi_{(2)}(x) + \ldots + \pi_{(c)}(x) \geq \tau\}, \qquad (3)$$

and the prediction set:

$$C_\alpha^{\text{oracle+}}(x) = \{`y` \text{ indices of the } L(x; \pi, 1 - \alpha) \text{ largest } \pi_y(x)\} . \qquad (4)$$

Hence, (4) is the smallest deterministic set that contains a response with feature values $X = x$ with probability at least $1 - \alpha$. For example, if $\pi_1(x) = 0.3$, $\pi_2(x) = 0.6$, and $\pi_3(x) = 0.1$, we have $\pi_{(1)}(x) = 0.6$, $\pi_{(2)}(x) = 0.3$, and $\pi_{(3)}(x) = 0.1$, with $L(x, 0.9) = 2$, $C_{0.1}^{\text{oracle}}(x) = \{1, 2\}$, and $L(x, 0.5) = 1$, $C_{0.5}^{\text{oracle}}(x) = \{2\}$. Furthermore, define a function $\mathcal{S}$ with input $x$, $u \in [0, 1]$, $\pi$, and $\tau$, which computes the set of most likely labels up to (but possibly excluding) the one identified by (3):

$$\mathcal{S}(x, u; \pi, \tau) = \begin{cases} `y` \text{ indices of the } L(x; \pi, \tau) - 1 \text{ largest } \pi_y(x), & \text{if } u \leq V(x; \pi, \tau), \\ `y` \text{ indices of the } L(x; \pi, \tau) \text{ largest } \pi_y(x), & \text{otherwise,} \end{cases} \qquad (5)$$

where

$$V(x; \pi, \tau) = \frac{1}{\pi_{(L(x;\pi,\tau))}(x)} \left[ \sum_{c=1}^{L(x;\pi,\tau)} \pi_{(c)}(x) - \tau \right].$$

With this in place, by letting $u$ be the realization of a uniform random variable, we can see that the oracle has access to tighter randomized prediction sets, namely,

$$C_\alpha^{\text{oracle}}(x) = \mathcal{S}(x, U; \pi, 1 - \alpha). \qquad (6)$$

Above, $U \sim \text{Uniform}(0, 1)$ is independent of everything else. It is easy to verify that the sets in (6) are the smallest randomized prediction sets with conditional coverage at level $1 - \alpha$. In the above example, we would have $C_{0.5}^{\text{oracle}}(x) = \emptyset$ with probability $(0.6 - 0.5)/0.6 = 1/6$ and $C_{0.5}^{\text{oracle}}(x) = \{2\}$ otherwise. Finally, if there are any ties among the class probabilities, the oracle could simply break them at random. Of course, we do not have access to such an oracle since $P_{Y|X}$ is unknown.

## 1.2 Preview of our methods

This paper uses classifiers trained on the available data to approximate the unknown conditional distribution of $Y \mid X$. A key strength of the proposed methods is their ability to work with any black-box predictive model, including neural networks, random forests, support vector classifiers, or any other currently existing or possible future alternatives. The only restriction on the training algorithm is that it should treat all samples exchangeably; i.e., it should be invariant to their order. Most off-the-shelf tools offer such suitable probability estimates $\hat{\pi}_y(x)$ that we can exploit, regardless of whether they are well-calibrated, by imputing them into an algorithm inspired by the oracle from Section 1.1 in order to obtain prediction sets with guaranteed coverage—as we shall see.

Our reader will understand that naively substituting $\pi_y(x)$ with $\hat{\pi}_y(x)$ into the oracle procedure would yield predictions lacking any statistical guarantees because $\hat{\pi}_y(x)$ may be a poor approximation of $\pi_y(x)$. Fortunately, we can automatically account for errors in $\hat{\pi}_y(x)$ by adaptively choosing the

threshold $\tau$ in (3) in such a way as to guarantee finite-sample coverage on future test points. The intuition is that setting $\tau = 1 - \alpha$ may not necessarily guarantee coverage at level $1 - \alpha$ for future test points, if $\hat{\pi} \neq \pi$. However, we can compute the empirical coverage on hold-out data as a function of $\tau$, and then select the smallest value of $\tau$ that leads to the desired $1 - \alpha$ coverage. Below, we will show that this adaptive tuning rigorously yields tight coverage.

## 1.3  Related work

We build upon conformal inference [12, 24, 26] and take inspiration from [3, 5, 8–10, 13, 18] which made conformal prediction for regression problems adaptive to heteroscedasticity, thus bringing it closer to conditional coverage [20]. Conformal inference has been applied before to classification problems [7, 19, 24, 25] in order to attain marginal coverage; however, the idea of explicitly trying to approximate the oracle from Section 1.1 is novel. We will see that our procedure empirically achieves better conditional coverage than a direct application of conformal inference. While working on this project, we became aware of the independent work of [2], which also seeks to improve the conditional coverage of conformal classification methods. However, their approach differs substantially; see Section 2.4. Finally, our method also naturally accommodates calibration through cross-validation+ and the jackknife+ [4], which had not yet been extended to classification, although the natural generality of these calibration techniques has also been very recently noted by others [10].

A different but related line of work focuses on post-processing the output of black-box classification algorithms to produce more accurate probability estimates [6, 11, 15, 16, 22, 27, 28], although without achieving prediction sets with provable finite-sample coverage. These techniques are complementary to our methods and may help further boost our performance by improving the accuracy of any given black box; however, we have not tested them empirically in this paper for space reasons.

## 2  Methods

### 2.1  Generalized inverse quantile conformity scores

Suppose we have a black-box classifier $\hat{\pi}_y(x)$ that estimates the true unknown class probabilities $\pi_y(x)$. Here, we only assume $\hat{\pi}_y(x)$ to be standardized: $0 \leq \hat{\pi}_y(x) \leq 1$, $\sum_{y=1}^{C} \hat{\pi}_y(x) = 1$, $\forall x, y$. An example may be the output of the softmax layer of a neural network, after normalization. In fact, almost any standard machine learning software, e.g., `sklearn`, can produce a suitable $\hat{\pi}$, either through random forests, k-nearest neighbors, or support vector machines, to name a few options. Then, we plug $\hat{\pi}$ into a modified version of the imaginary oracle procedure of Section 1.1 where the threshold $\tau$ needs to be carefully calibrated using hold-out samples independent of the training data. We will present two alternative methods for calibrating $\tau$; both are based on the following idea.

We define a function $E$, with input $x, y, u, \hat{\pi}$, which computes the smallest value of $\tau$ such that the set $\mathcal{S}(x, u; \hat{\pi}, \tau)$ in (5) contains the label $y$ conditional on $X = x$. We call this the *generalized inverse quantile* conformity score function:

$$E(x, y, u; \hat{\pi}) = \min \left\{ \tau \in [0, 1] : y \in \mathcal{S}(x, u; \hat{\pi}, \tau) \right\}. \tag{7}$$

By construction, our scores evaluated on hold-out samples $(X_i, Y_i)$, namely $E_i = E(X_i, Y_i, U_i; \hat{\pi})$, are uniformly distributed conditional on $X$ if $\hat{\pi} = \pi$. (Each $U_i$ is a uniform random variable in $[0, 1]$ independent of everything else.) Therefore, one could also intuitively look at (7) as a special type of *p-value*. It is worth emphasizing that this property makes our scores naturally comparable across different samples, in contrast with the scores found in the earlier literature on adaptive conformal inference [18]. In fact, alternative conformity scores [2, 10, 12, 18] generally have different distributions at different values of $X$, even in the ideal case where the base method (our $\hat{\pi}$) is a perfect oracle. Below, we shall see that, loosely speaking, we can construct prediction sets with provable marginal coverage for future test points by applying (5) with a value of $\tau$ close to the $1 - \alpha$ quantile of $\{E_i\}_{i \in \mathcal{I}_2}$, where $\mathcal{I}_2$ is the set of hold-out data points not used to train $\hat{\pi}$; see (8).

### 2.2  Adaptive classification with split-conformal calibration

Algorithm 1 implements the above idea with split-conformal calibration, from which we begin because it is the easiest to explain. Later, we will consider alternative calibration methods based on

cross-validation+ and the jackknife+; we do not discuss full-conformal calibration in the interest of space, and because it is often computationally prohibitive. For simplicity, we will apply Algorithm 1 by splitting the data into two sets of equal size; however, this is not necessary and using more data points for training may sometimes perform better in practice [20].

---

**Algorithm 1:** Adaptive classification with split-conformal calibration

---

1 **Input:** data $\{(X_i, Y_i)\}_{i=1}^n$, $X_{n+1}$, black-box learning algorithm $\mathcal{B}$, level $\alpha \in (0, 1)$.
2 Randomly split the training data into 2 subsets, $\mathcal{I}_1, \mathcal{I}_2$.
3 Sample $U_i \sim \text{Uniform}(0, 1)$ for each $i \in \{1, \dots, n+1\}$, independently of everything else.
4 Train $\mathcal{B}$ on all samples in $\mathcal{I}_1$: $\hat{\pi} \leftarrow \mathcal{B}(\{(X_i, Y_i)\}_{i \in \mathcal{I}_1})$.
5 Compute $E_i = E(X_i, Y_i, U_i; \hat{\pi})$ for each $i \in \mathcal{I}_2$, with the function $E$ defined in (7).
6 Compute $\hat{Q}_{1-\alpha}(\{E_i\}_{i \in \mathcal{I}_2})$ as the $\lceil(1-\alpha)(1+|\mathcal{I}_2|)\rceil$th largest value in $\{E_i\}_{i \in \mathcal{I}_2}$.
7 Use the function $\mathcal{S}$ defined in (5) to construct the prediction set at $X_{n+1}$ as:

$$\hat{\mathcal{C}}_{n,\alpha}^{\text{SC}}(X_{n+1}) = \mathcal{S}(X_{n+1}, U_{n+1}; \hat{\pi}, \hat{Q}_{1-\alpha}(\{E_i\}_{i \in \mathcal{I}_2})). \tag{8}$$

8 **Output:** A prediction set $\hat{\mathcal{C}}_{n,\alpha}^{\text{SC}}(X_{n+1})$ for the unobserved label $Y_{n+1}$.

---

**Theorem 1.** *If the samples $(X_i, Y_i)$, for $i \in \{1, \dots, n+1\}$, are exchangeable and $\mathcal{B}$ from Algorithm 1 is invariant to permutations of its input samples, the output of Algorithm 1 satisfies:*

$$\mathbb{P}\left[Y_{n+1} \in \hat{\mathcal{C}}_{n,\alpha}^{\text{SC}}(X_{n+1})\right] \geq 1 - \alpha. \tag{9}$$

*Furthermore, if the scores $E_i$ are almost surely distinct, the marginal coverage is near tight:*

$$\mathbb{P}\left[Y_{n+1} \in \hat{\mathcal{C}}_{n,\alpha}^{\text{SC}}(X_{n+1})\right] \leq 1 - \alpha + \frac{1}{|\mathcal{I}_2| + 1}. \tag{10}$$

The proofs of this theorem and all other results are in Supplementary Section S2. Marginal coverage holds regardless of the quality of the black-box approximation; however, one can intuitively expect that if the black-box is consistent and a large amount of data is available, so that $\hat{\pi}_y(x) \approx \pi_y(x)$, the output of our procedure will tend to be a close approximation of the output of the oracle, which provides optimal conditional coverage. This statement could be made rigorous under some additional technical assumptions besides the consistency of the black box [20]. However, we prefer to avoid tedious technical details, especially since the intuition is already clear. If $\hat{\pi} = \pi$, the sets $\mathcal{S}(X_i, U_i; \pi, \tau)$ in (5) will tend to contain the true labels for a fraction $\tau$ of the points $i \in \mathcal{I}_2$, as long as $|\mathcal{I}_2|$ is large. In this limit, $\hat{Q}_{1-\alpha}(\{E_i\}_{i \in \mathcal{I}_2})$ becomes approximately equal to $1 - \alpha$, and the predictions in (8) will eventually approach those in (6).

### 2.3 Adaptive classification with cross-validation+ and jackknife+ calibration

A limitation of Algorithm 1 is that it only uses part of the data to train the predictive algorithm. Consequently, the estimate $\hat{\pi}$ may not be as accurate as it could have been had we used all the data for estimation purposes. This is especially true if the sample size $n$ is small. Algorithm 2 presents an alternative solution that replaces data splitting with a cross-validation approach, which is computationally more expensive but often provides tighter prediction sets.

In words, in Algorithm 2, we sweep over all possible labels $y \in \mathcal{Y}$ and include $y$ in the final prediction set $\hat{\mathcal{C}}_{n,\alpha}^{\text{CV+}}(X_{n+1})$ if the corresponding score $E(X_{n+1}, y, U_{n+1}; \hat{\pi}^{k(i)})$ is smaller than $(1-\alpha)(n+1)$ hold-out scores $E(X_i, Y_i, U_i; \hat{\pi}^{k(i)})$ evaluated on the true labeled data. Note that we have assumed $n/K$ to be an integer for simplicity; however, different splits can have different sizes. In the special case where $K = n$, we refer to the hold-out system in Algorithm 2 as jackknife+ rather than cross-validation+, consistently with the terminology in [4].

**Theorem 2.** *Under the same assumptions of Theorem 1, the output of Algorithm 2 satisfies:*

$$\mathbb{P}\left[\{Y_{n+1} \in \hat{\mathcal{C}}_{n,\alpha}^{\text{CV+}}(X_{n+1})\right] \geq 1 - 2\alpha - \min\left\{\frac{2(1-1/K)}{n/K+1}, \frac{1-K/n}{K+1}\right\}. \tag{12}$$

*In the special case where $K = n$, this bound simplifies to:*

$$\mathbb{P}\left[Y_{n+1} \in \hat{\mathcal{C}}_{n,\alpha}^{\text{JK+}}(X_{n+1})\right] \geq 1 - 2\alpha. \tag{13}$$

---

**Algorithm 2:** Adaptive classification with CV+ calibration

---

1 **Input:** data $\{(X_i, Y_i)\}_{i=1}^n$, $X_{n+1}$, black-box $\mathcal{B}$, number of splits $K \leq n$, level $\alpha \in (0, 1)$.
2 Randomly split the training data into $K$ disjoint subsets, $\mathcal{I}_1, \ldots, \mathcal{I}_K$, each of size $n/K$.
3 Sample $U_i \sim \text{Uniform}(0, 1)$ for each $i \in \{1, \ldots, n+1\}$, independently of everything else.
4 **for** $k \in \{1, \ldots, K\}$ **do**
5 $\quad$ Train $\mathcal{B}$ on all samples except those in $\mathcal{I}_k$: $\hat{\pi}^k \leftarrow \mathcal{B}(\{(X_i, Y_i)\}_{i \in \{1, \ldots, n\} \setminus \mathcal{I}_k})$.
6 **end**
7 Use the function $E$ defined in (7) to construct the prediction set $\hat{\mathcal{C}}_{n,\alpha}^{\text{CV+}}(X_{n+1})$ as:

$$\hat{\mathcal{C}}_{n,\alpha}^{\text{CV+}}(X_{n+1}) = \Bigg\{ y \in \mathcal{Y} :$$
$$\sum_{i=1}^n \mathbf{1}\left[ E(X_i, Y_i, U_i; \hat{\pi}^{k(i)}) < E(X_{n+1}, y, U_{n+1}; \hat{\pi}^{k(i)}) \right] < (1-\alpha)(n+1) \Bigg\}, \quad (11)$$

$\quad$ where $k(i) \in \{1, \ldots, K\}$ is the fold containing the $i$th sample.
8 **Output:** A prediction set $\hat{\mathcal{C}}_{n,\alpha}^{\text{CV+}}(X_{n+1})$ for the unobserved label $Y_{n+1}$.

---

Note that this establishes that the coverage is slightly below $1 - 2\alpha$. Therefore, to guarantee $1 - \alpha$ coverage, we should replace the input $\alpha$ in Algorithm 2 with a smaller value near $\alpha/2$. We chose not to do so because our experiments show that the current implementation already typically covers at level $1 - \alpha$ (or even higher) in practice; this empirical observation is consistent with [4]. Furthermore, there exists a conservative variation of Algorithm 2 for which we can prove $1 - \alpha$ coverage without modifying the input level; see Supplementary Section S1.1.

To see why everything above makes sense, consider what would happen if the black-box estimates of conditional probabilities in Algorithm 2 were exact. In this case, the final prediction set in (11) would become

$$\hat{\mathcal{C}}_{n,\alpha}^{\text{CV+}}(X_{n+1}) = \left\{ y \in \mathcal{Y} : E(X_{n+1}, y, U_{n+1}; \pi) < \hat{Q}_{1-\alpha}(\{E(X_i, Y_i, U_i; \pi)\}_{i \in \{1, \ldots, n\}}) \right\}, \quad (14)$$

where $\hat{Q}_{1-\alpha}$ is defined as in Section 2.2. If $n$ is large, for any fixed threshold $\tau$, we can expect $\mathcal{S}(X_i, U_i; \pi, \tau)$ to contain $Y_i$ for approximately a fraction $\tau$ of samples $i$. Therefore, $\hat{Q}_{1-\alpha}(\{E(X_i, Y_i, U_i; \pi)\}_{i \in \{1, \ldots, n\}}) \approx 1 - \alpha$, and the decision rule becomes approximately:

$$\hat{\mathcal{C}}_{n,\alpha}^{\text{CV+}}(X_{n+1}) \approx \{y \in \mathcal{Y} : E(X_{n+1}, y, U_{n+1}; \pi) \leq 1 - \alpha\}, \quad (15)$$

which is equivalent to the oracle procedure from Section 1.1.

## 2.4 Comparison with alternative conformal methods

Conformal prediction has been proposed before in the context of classification [24], through a very general calibration rule of the form

$$\hat{\mathcal{C}}(x; t) = \{y \in \mathcal{Y} : \hat{f}(y \mid x) \geq t\},$$

where the score $\hat{f}$ is a function learned by a black-box classifier. However, to date it was not clear how to best translate the output of the classifier into a powerful score $\hat{f}$ for the above decision rule. In fact, typical choices of $\hat{f}(y \mid x)$, e.g., the estimated probability of $Y = y$ given $X = x$, often lead to poor conditional coverage because the same threshold $t$ is applied both to easy-to-classify samples (where one label has probability close to 1 given $X$) and to hard-to-classify samples (where all probabilities are close to $1/|\mathcal{Y}|$ given $X$). Therefore, this *homogeneous* conformal classification may significantly underperform compared to the oracle from Section 1.1, even in the ideal case where the black-box manages to learn the correct probabilities. This limitation has also been very recently noted in [2] and is analogous to that addressed by [18] in problems with a continuous response variable [20].

The work of [2] addresses this problem by applying quantile regression [18] to hold-out scores $\hat{f}$. However, their solution has two limitations. Firstly, it involves additional data splitting to avoid

overfitting, which tends to reduce power. Secondly, its theoretical asymptotic optimality is weaker than ours because it requires the consistency of two black-boxes instead of one (this should be clear even though we have explained consistency only heuristically). Practically, experiments suggest that our method provides superior conditional coverage and often yields smaller prediction sets.

## 2.5 Extension to label-conditional coverage

Our method can be easily extended to obtain provable coverage separately for each class [19, 24]:

$$\mathbb{P}\left[Y_{n+1} \in \hat{\mathcal{C}}_{n,\alpha}(X_{n+1}) \mid Y_{n+1} = y\right] \geq 1 - \alpha, \qquad \forall y \in \mathcal{Y}. \tag{16}$$

The only difference is that the threshold $\tau$ should be calibrated separately for each class. More precisely, focusing on the extension of Algorithm 1 for simplicity, we would compute $\hat{Q}^{(y)}_{1-\alpha}(\{E_i\}_{i \in \mathcal{I}_2})$ as the $\lceil (1-\alpha)(1 + |\{i \in \mathcal{I}_2 : Y_i = y\}|)\rceil$th largest value in $\{E_i\}_{i \in \mathcal{I}_2 : Y_i = y}$, for each $y \in \mathcal{Y}$. Then, we would define $\hat{\tau} = \max_{y \in \mathcal{Y}} \hat{Q}^{(y)}_{1-\alpha}(\{E_i\}_{i \in \mathcal{I}_2})$ and output $\hat{\mathcal{C}}^{\mathrm{SC-lc}}_{n,\alpha}(X_{n+1}) = \mathcal{S}(X_{n+1}, U_{n+1}; \hat{\pi}, \hat{\tau})$. More details about this extension are in Supplementary Section S1.2.

In the interest of simplicity, we have not explicitly sought label-conditional coverage in the following numerical experiments. Nonetheless, we shall see from the results in Figure 3 that our method performs relatively well in terms of label-conditional coverage even without the explicit constraint.

# 3 Experiments with simulated data

## 3.1 Methods and metrics

We compare the performances of Algorithms 1 (SC) and 2 (CV+, JK+), which are based on the new generalized inverse quantile conformity scores in (7), to those of homogeneous conformal classification (HCC) and conformal quantile classification (CQC) [2]. We focus on two different data generating scenarios in which marginal coverage is not a good proxy for conditional coverage (the second setting is discussed in Supplementary Section S3.3). In both cases, we explore 3 different black-boxes: an *oracle* that knows the true $\pi_y(x)$ for all $y \in \mathcal{Y}$ and $x$; a support vector classifier (SVC) implemented by the `sklearn` Python package; and a random forest classifier (RFC) also implemented by `sklearn`— see Supplementary Section S3.1 for more details.

We fix $\alpha = 0.1$ and assess performance in terms of marginal coverage, conditional coverage, and mean cardinality of the prediction sets. Conditional coverage is defined using an estimate of the worst-slice (WS) coverage similar to that in [2], as explained in Supplementary Section S1.3. The cardinality of the prediction sets is computed both marginally and conditionally on coverage; the former is defined as $\mathbb{E}[|\hat{\mathcal{C}}(X_{n+1})|]$ and the latter as $\mathbb{E}[|\hat{\mathcal{C}}(X_{n+1})| \mid Y_{n+1} \in \hat{\mathcal{C}}(X_{n+1})]$. Additional coverage and size metrics defined by conditioning on the value of a given discrete feature, e.g., $X_1$, are discussed in Supplementary Section S3.

## 3.2 Experiments with multinomial model and inhomogeneous features

We generate the features $X \in \mathbb{R}^p$, with $p = 10$, as follows: $X_1 = 1$ w.p. $1/5$ and $X_1 = -8$ otherwise, while $X_2, \ldots, X_{10}$ are independent standard normal. The conditional distribution of $Y \in \{1, \ldots, 10\}$ given $X = x$ is multinomial with weights $w_j(x)$ defined as $w_j(x) = z_j(x)/\sum_{j'=1}^p z_{j'}(x)$, where $z_j(x) = \exp(x^T \beta_j)$ and each $\beta_j \in \mathbb{R}^p$ is sampled from an independent standard normal distribution.

Figure 1 confirms that our methods have valid conditional coverage if the true class probabilities are provided by an oracle. If the probabilities are estimated by the RFC, the conditional coverage appears to be only slightly below $1 - \alpha$, and is near perfect with the SVC black box. By contrast, the conditional coverage of the alternative methods is always significantly lower than $1 - \alpha$, even with the help of the oracle. Our methods produce slightly larger prediction sets when the oracle is available, but our sets are typically smaller than those of CQC and only slightly larger than those of HCC when the class probabilities are estimated. Finally, note that JK+ is the most powerful of our methods, followed by CV+, although SC is computationally more affordable.

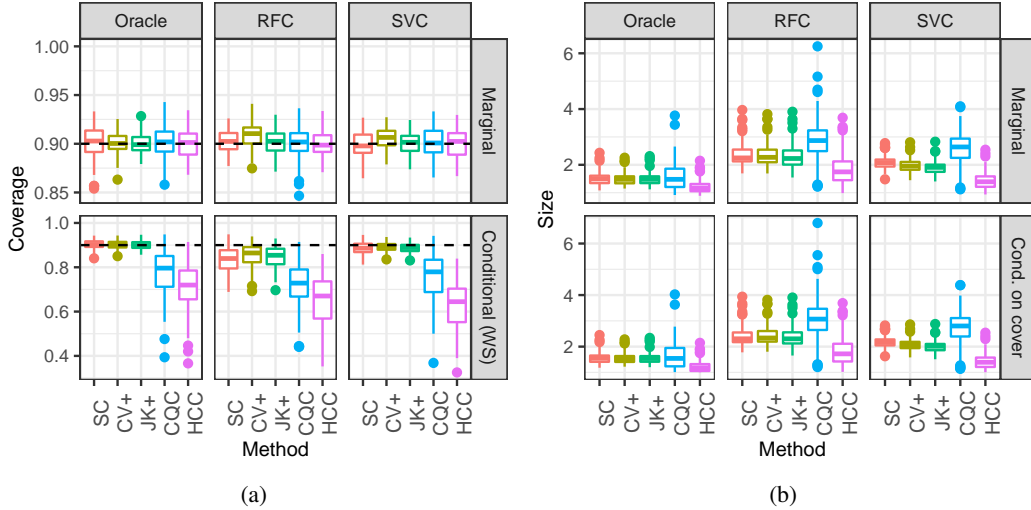

Figure 1: Several classification methods on simulated data with 10 classes, for different choices of calibration and black-box models. SC, CV+, and JK+ are applied with our new generalized inverse quantile conformity scores defined in (7). The results correspond to 100 independent experiments with 1000 training samples and 5000 test samples each. All methods have 90% marginal coverage. (a): Marginal coverage and worst-slice conditional coverage. (b): Size of prediction sets.

## 4 Experiments with real data

In this section, we compare the performance of our proposed methods (SC, CV+, and JK+) with the new generalized inverse quantile conformity scores defined in (7) to those of HCC and CQC [2]. We found that the original suggestion of [2] to fit a quantile neural network [21] on the class probability score can be unstable and yield very wide predictions. Therefore, we offer a second variant of this calibration method, denoted by CQC-RF, which replaces the quantile neural network estimator with quantile random forests [14]; see Supplementary Section S4 for details.

The validity and statistical efficiency of each method is evaluated according to the same metrics as in Section 3. In all experiments, we set $\alpha = 0.1$ and use the following base predictive models: (i) kernel SVC, (ii) random forest classifier (RFC), and (iii) two-layer neural network classifier (NNet). A detailed description of each algorithm and corresponding hyper-parameters is in Supplementary Section S4. The methods are tested on two well-known data sets: the Mice Protein Expression data set[3] and the MNIST handwritten digit data set. The supplementary material describes the processing pipeline and discusses additional experiments on the Fashion-MNIST and CIFAR10 data sets. Supplementary Tables S1–S4 summarize the results of our experiments in more detail and also consider additional settings.

Figure 2 shows that all methods attain valid marginal coverage on the Mice Protein Expression data, as expected. Here, HCC, CQC, and CQC-RF fail to achieve conditional coverage, in contrast to the proposed methods (SC, CV+, JK+) based on our new conformity scores in (7). Turning to efficiency, we observe that the prediction sets of CV+ and JK+ are smaller than those of SC, and comparable in size to those of HCC. Here, the original CQC algorithm performs poorly both in terms of conditional coverage and cardinality. The CQC-RF variant is not as unstable as the original CQC, although it does not perform much better than HCC.

Figure 3 presents the results on the MNIST data. Here, the sample size is relatively large and hence we exclude JK+ due to its higher computational cost. As in the previous experiments, all methods achieve 90% marginal coverage. Unlike CQC, CQC-RF, and HCC, our methods also attain valid conditional coverage when relying on the NNet or SVC as base predictive models. With the RFC, all methods tend to undercover, suggesting that this classifier estimates the class probabilities poorly, and our prediction sets are larger than those constructed by CQC-RF and HCC. By contrast, the

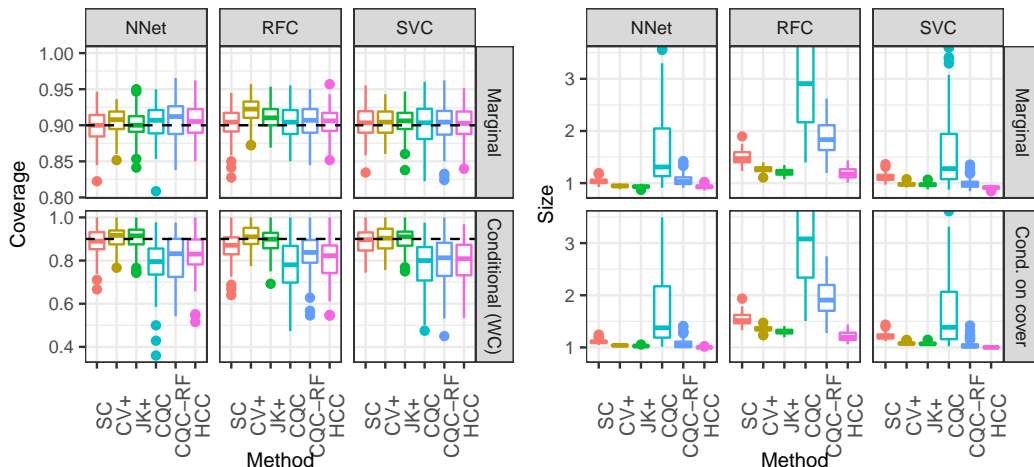

Figure 2: Experiments on Mice Protein Expression data. 100 independent experiments with 500 randomly chosen training samples and 580 test samples each. Left: coverage. Right: size of prediction sets (extremely large values for CQC not shown). Other details are as in Figure 1.

NNet enables our methods to achieve conditional coverage with prediction sets comparable in size to those produced by CQC-RF and HCC. The bottom part of Figure 3 demonstrates that CV+ also has conditional coverage given the true class label $Y$, while SC performs only slightly worse. In striking contrast, both HCC, CQC, and CQC-RF fail to achieve 90% conditional coverage.

## 5  Conclusions

This paper introduced a principled and versatile modular method for constructing prediction sets for multi-class classification problems that enjoy provable finite-sample coverage, and also behave well in terms of conditional coverage when compared to alternatives. Our approach leverages the power of any black-box machine learning classifier that may be available to practitioners, and is easily calibrated via various hold-out procedures; e.g., conformal splitting, CV+, or the jackknife+. This flexibility makes our approach widely applicable and offers options to balance between computational efficiency, data parsimony, and power.

Although this paper focused on classification, using conformity scores similar to those in (7) to calibrate hold-out procedures for regression problems [4, 18] is tantalizing. In fact, previous work in the regression setting focused on conformity scores that measure the distance of a data point from its predicted interval on the scale of the $Y$ values (which makes sense for homoscedastic regression, but may not be optimal otherwise), rather than by the amount one would need to relax the nominal threshold (our $\tau$) until the true value is covered. We leave it to future work to explore the performance of our intuitive metrics in other settings.

The Python package at `https://github.com/msesia/arc` implements our methods. This repository also contains code to reproduce our experiments.

## Broader Impact

Machine learning algorithms are increasingly relied upon by decision makers. It is therefore crucial to combine the predictive performance of such complex machinery with practical guarantees on the reliability and uncertainty of their output. We view the calibration methods presented in this paper as an important step towards this goal. In fact, uncertainty estimation is an effective way to quantify and communicate the benefits and limitations of machine learning. Moreover, the proposed methodologies provide an attractive way to move beyond the standard prediction accuracy measure used to compare algorithms. For instance, one can compare the performance of two candidate predictors, e.g., random forest and neural network (see Figure 3), by looking at the size of the corresponding prediction sets

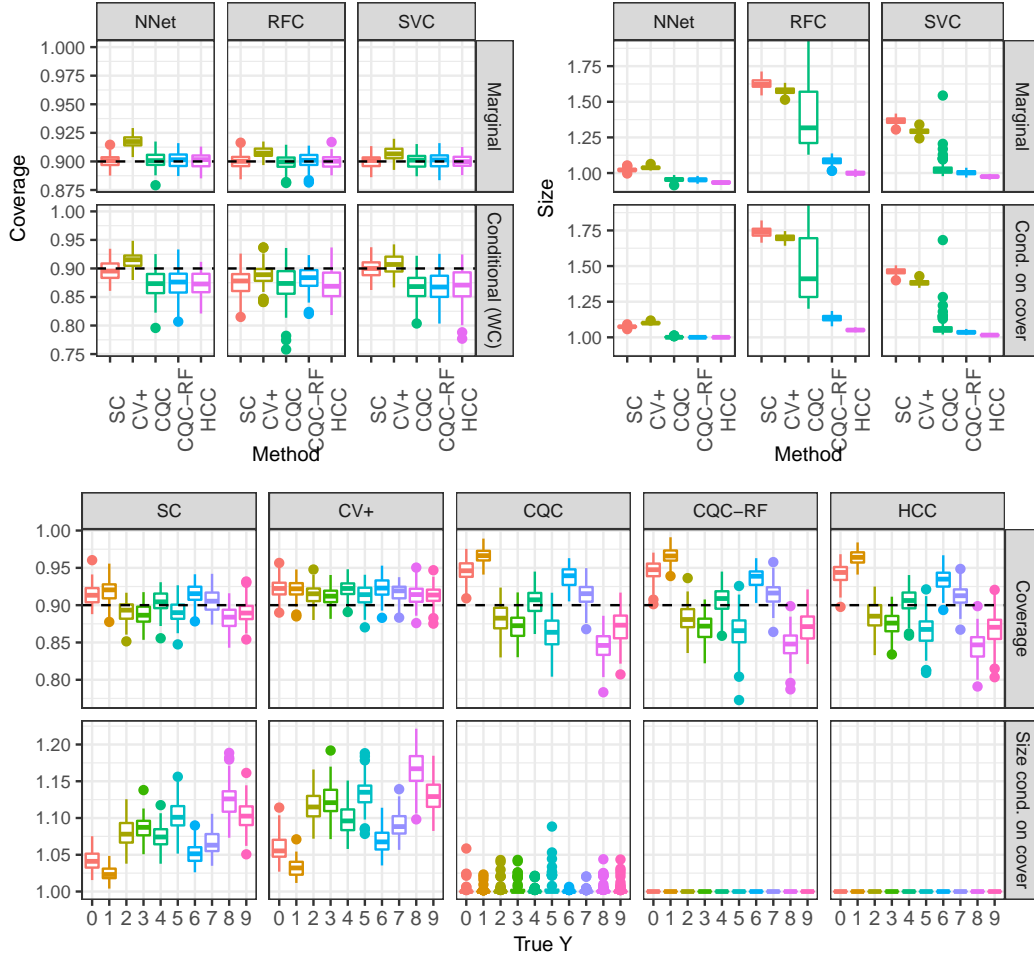

Figure 3: Experiments on MNIST data. 100 independent experiments with 10000 randomly chosen training samples and 5000 test samples each. Top: coverage and size of prediction sets (large values for CQC not shown). Bottom: coverage with neural network black box, conditional on the true $Y$, and size of the corresponding prediction sets, conditional on coverage. Other details are as in Figure 1.

and/or their their conditional coverage. Finally, the approximate conditional coverage that we seek in this work is highly relevant within the broader framework of fairness, as discussed by [17] within a regression setting. While our approximate conditional coverage already implicitly reduces the risk of unwanted bias, an equalized coverage requirement [17] can also be easily incorporated into our methods to explicitly avoid discrimination based on protected categories.

We conclude by emphasizing that the validity of our methods relies on the exchangeability of the data points. If this assumption is violated (e.g., with time-series data), our prediction sets may not have the right coverage. A general suggestion here is to always try to leverage specific knowledge of the data and of the application domain to judge whether the exchangeability assumption is reasonable. Finally, our data-splitting techniques in Section 4 offer a practical way to verify empirically the validity of the predictions on any given data set.

## Acknowledgments and Disclosure of Funding

E. C. was partially supported by Office of Naval Research grant N00014-20-12157, and by the Army Research Office (ARO) under grant W911NF-17-1-0304. Y. R. was partially supported by ARO under the same grant. Y. R. thanks the Zuckerman Institute, ISEF Foundation, the Viterbi Fellowship,

Technion, and the Koret Foundation, for providing additional research support. M. S. was suported by NSF grant DMS 1712800. Y. R. and M. S. were advised by E. C. at Stanford University.

## Footnotes

[2]Recall that the conditional quantiles for continuous responses are: $\inf\{y \in \mathbb{R} : \mathbb{P}[Y \leq y \mid X = x] \geq \tau\}$.

[3]https://archive.ics.uci.edu/ml/datasets/Mice+Protein+Expression

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
