[Supplementary Material]

# Supplementary Material for
# Classification with Valid and Adaptive Coverage

**Yaniv Romano**[*]
Department of Statistics
Stanford University
Stanford, CA, USA
yromano@stanford.edu

**Matteo Sesia**[*]
Department of Data Sciences and Operations
University of Southern California
Los Angeles, CA, USA
sesia@marshall.usc.edu

**Emmanuel J. Candès**
Departments of Mathematics and of Statistics
Stanford University
Stanford, CA, USA
candes@stanford.edu

## S1  Supplementary methods

### S1.1  Adaptive classification with minimax jackknife+ calibration

We can apply the minimax calibration technique of [2] to obtain a non-trivial variation of Algorithm 2 for which marginal coverage can be rigorously proved at level $1 - \alpha$, without modifying the current input level. Here, we consider the jackknife+—i.e., $K = n$—for simplicity. The only difference with Algorithm 2 is that the prediction set in (11) is replaced by the following larger set:

$$
\hat{\mathcal{C}}_{n,\alpha}^{\text{J+mm}}(X_{n+1}) = \Big\{ y \in \mathcal{Y} :
$$
$$
\sum_{i=1}^{n} \mathbf{1}\left[ E(X_i, Y_i, U_i; \hat{\pi}^i) < \min_{j \in \{1,\dots,n\}} E(X_{n+1}, y, U_{n+1}; \hat{\pi}^j) \right] < (1-\alpha)(n+1) \Big\}. \tag{S1}
$$

**Theorem S1.** *Under the same assumptions of Theorem 1, the output of Algorithm 2 with $K = n$, and (11) replaced by (S1), satisfies:*

$$
\mathbb{P}\left[ Y_{n+1} \in \hat{\mathcal{C}}_{n,\alpha}^{\text{J+mm}}(X_{n+1}) \right] \geq 1 - \alpha. \tag{S2}
$$

### S1.2  Adaptive classification with label-conditional coverage

Algorithm S1 describes the extension of Algorithm 1 discussed in Section 2.5, which ensures label-conditional coverage. The validity of this algorithm is established by the following result.

**Theorem S2.** *If the samples $(X_i, Y_i)$, for $i \in \{1, \dots, n+1\}$, are exchangeable and $\mathcal{B}$ from Algorithm S1 is invariant to permutations of its input samples, the output of Algorithm S1 satisfies:*

$$
\mathbb{P}\left[ Y_{n+1} \in \hat{\mathcal{C}}_{n,\alpha}^{\text{SC}-\text{lc}}(X_{n+1}) \mid Y_{n+1} = y \right] \geq 1 - \alpha, \tag{S4}
$$

*for all $y \in \mathcal{Y}$.*

---

[*]Equal contribution.

**Algorithm S1:** Split-conformal adaptive classification with label-conditional coverage

1 **Input:** data $\{(X_i, Y_i)\}_{i=1}^n$, $X_{n+1}$, black-box learning algorithm $\mathcal{B}$, level $\alpha \in (0,1)$,
2      list of possible labels $\mathcal{Y}$.
3 Randomly split the training data into 2 subsets, $\mathcal{I}_1, \mathcal{I}_2$.
4 Sample $U_i \sim \text{Uniform}(0,1)$ for each $i \in \{1, \ldots, n+1\}$, independently of everything else.
5 Train $\mathcal{B}$ on all samples in $\mathcal{I}_1$: $\hat{\pi} \leftarrow \mathcal{B}(\{(X_i, Y_i)\}_{i \in \mathcal{I}_1})$.
6 Compute $E_i = E(X_i, Y_i, U_i; \hat{\pi})$ for each $i \in \mathcal{I}_2$, with the function $E$ defined in (7).
7 **for** $y \in \mathcal{Y}$ **do**
8    Compute $\hat{Q}_{1-\alpha}^{(y)}(\{E_i\}_{i \in \mathcal{I}_2})$ as the $\lceil (1-\alpha)(1 + |\{i \in \mathcal{I}_2 : Y_i = y\}|) \rceil$th largest value in $\{E_i\}_{i \in \mathcal{I}_2 : Y_i = y}$.
9 **end**
10 Compute $\hat{\tau} = \max_{y \in \mathcal{Y}} \hat{Q}_{1-\alpha}^{(y)}(\{E_i\}_{i \in \mathcal{I}_2})$.
11 Use the function $\mathcal{S}$ defined in (5) to construct the prediction set at $X_{n+1}$ as:

$$\hat{\mathcal{C}}_{n,\alpha}^{\text{SC-lc}}(X_{n+1}) = \mathcal{S}(X_{n+1}, U_{n+1}; \hat{\pi}, \hat{\tau}). \tag{S3}$$

12 **Output:** A prediction set $\hat{\mathcal{C}}_{n,\alpha}^{\text{SC-lc}}(X_{n+1})$ for the unobserved label $Y_{n+1}$.

### S1.3 Quantifying conditional coverage in finite samples

Similarly to the approach of [1], we measure coverage over a slab

$$S_{v,a,b} = \{x \in \mathbb{R}^p : a \leq v^T x \leq b\}$$

of the feature space, where the values of $v \in \mathbb{R}^p$ and $a < b \in \mathbb{R}$ are chosen adversarially but independently of the data. In particular, for any fixed classification prediction set $\hat{\mathcal{C}}$ and $\delta \in (0,1)$, we define

$$\text{WSC}(\hat{\mathcal{C}}; \delta) = \inf_{v \in \mathbb{R}^p, \, a < b \in \mathbb{R}} \left\{ \mathbb{P}[Y \in \hat{\mathcal{C}}(X) \mid X \in S_{v,a,b}] \text{ s.t. } \mathbb{P}[X \in S_{v,a,b}] \geq 1 - \delta] \right\}.$$

In practice, we estimate WSC for a particular $\hat{\mathcal{C}}$ by sampling 1000 independent vectors $v$ on the unit sphere in $\mathbb{R}^p$ and optimizing the corresponding parameters $a, b$ through a grid search. To avoid finite-sample negative bias, we partition the test data into two subsets (e.g., containing 25% and 75% of the samples respectively); then, we use the first subset to estimate the optimal values $v^*, a^*, b^*$, and the second subset to evaluate conditional coverage:

$$\mathbb{P}[Y \in \hat{\mathcal{C}}(X) \mid X \in S_{v^*, a^*, b^*}]. \tag{S5}$$

Therefore, regardless of the quality of our solution $v^*, a^*, b^*$ to the above optimization problem, the quantity in (S5) should be equal to the nominal coverage level $1 - \alpha$ for any method with valid conditional coverage. However, it is worth highlighting that controlling (S5) does not necessarily imply that conditional coverage holds more generally, which is why we also look at alternative measures of conditional coverage given either the value of certain features (e.g., $X_1$), or that of the true label $Y$.

## S2 Supplementary proofs

*Proof of Theorem 1.* We begin by proving the lower bound on coverage. By construction of the prediction set in (8), we know that

$$Y_{n+1} \in \hat{\mathcal{C}}_{n,\alpha}^{\text{SC}}(X_{n+1})$$

if and only if

$$\min \left\{ \tau \in [0,1] : Y_{n+1} \in \mathcal{S}(X_{n+1}, U_{n+1}; \hat{\pi}, \tau) \right\} \leq \hat{Q}_{1-\alpha}(\{E_i\}_{i \in \mathcal{I}_2}),$$

or, equivalently, if and only if

$$E_{n+1} \leq \hat{Q}_{1-\alpha}(\{E_i\}_{i \in \mathcal{I}_2}). \tag{S6}$$

Since all the conformity scores $E_{n+1}$ and $\{E_i\}_{i \in \mathcal{I}_2}$ are exchangeable, the probability of the event in (S6) can be no smaller than $1 - \alpha$. The formal proof of this statement is standard at this point, so we simply refer to [3] for the remaining technical details. The proof for the upper bound also immediately follows from (S6) by applying Lemma 2 in [3]. $\qquad\square$

*Proof of Theorem 2.* The proof is essentially an application of the main result in [2]. This will become apparent after we reduce our claim to the setting in the aforementioned paper. We now examine this reduction.

Imagine that we have access to $m = n/K$ test points

$$(X_{n+1}, Y_{n+1}, U_{n+1}), \ldots, (X_{n+m}, Y_{n+m}, U_{n+m})$$

as well as the training data; we will call this data set the *augmented* data set. After partitioning the training data into sets $\mathcal{I}_1, \ldots, \mathcal{I}_K$ of size $m$, we define $\mathcal{I}_{K+1} = \{n+1, \ldots, n+m\}$ as the set of test points. For any distinct $k, k' \in \{1, \ldots, K+1\}$, let $\tilde{\pi}^{k,k'}$ define the class probability estimator obtained by fitting the black box on the data in $\{1, \ldots, n+m\} \setminus (\mathcal{I}_k \cup \mathcal{I}_{k'})$. Note that $\tilde{\pi}^{k,K+1} = \hat{\pi}^k$ for any $k$.

Next, define the matrix $R \in \mathbb{R}^{(n+m) \times (n+m)}$ with entries

$$R_{i,j} = \begin{cases} +\infty, & \text{if } k(i) = k(j), \\ E(X_i, Y_i, U_i; \tilde{\pi}^{k(i),k(j)}), & \text{if } k(i) \neq k(j), \end{cases}$$

and the comparison matrix $A \in \{0,1\}^{(n+m) \times (n+m)}$ with entries

$$A_{ij} = \mathbf{1}\left[R_{ij} > R_{ji}\right]. \tag{S7}$$

Note that

$$Y_{n+1} \notin \hat{\mathcal{C}}^{\mathrm{CV+}}_{n,\alpha}(X_{n+1}) \iff (n+1) \in \mathcal{F}(A),$$

where the set $\mathcal{F}(A)$ is defined as in [2]:

$$\mathcal{F}(A) = \left\{ i \in \{1, \ldots, n+m\} : \sum_{j=1}^{n+m} A_{i,j} \geq (1-\alpha)(n+1) \right\}. \tag{S8}$$

The rest of the proof follows directly by applying Lemma S1 below, which is established by the proof of Theorem 4 in [2]. To invoke this lemma, we only need to check that $A \stackrel{d}{=} \Sigma A \Sigma^\top$, where $A$ is defined as in (S7), and $\Sigma$ is any permutation matrix that does not mix points assigned to different folds (so that the unordered set of probability estimators $\{\hat{\pi}^k\}_{k=1}^K$ is invariant). This is easy to verify. Let $\sigma(1), \ldots, \sigma(n+m)$ be the permutation of the data points corresponding to $\Sigma$, so that

$$(\Sigma A \Sigma^\top)_{ij} = A_{\sigma(i)\sigma(j)}.$$

Then, for any $i, j$ such that $k(i) \neq k(j)$,

$$
\begin{aligned}
A_{\sigma(i)\sigma(j)} &= \mathbf{1}\left[E(X_{\sigma(i)}, Y_{\sigma(i)}, U_{\sigma(i)}; \tilde{\pi}^{k(\sigma(i)),k(\sigma(j))}) > E(X_{\sigma(j)}, Y_{\sigma(j)}, U_{\sigma(j)}; \tilde{\pi}^{k(\sigma(i)),k(\sigma(j))})\right] \\
&= \mathbf{1}\left[E(X_{\sigma(i)}, Y_{\sigma(i)}, U_{\sigma(i)}; \tilde{\pi}^{k(i),k(j)}) > E(X_{\sigma(j)}, Y_{\sigma(j)}, U_{\sigma(j)}; \tilde{\pi}^{k(i),k(j)})\right] \\
&\stackrel{d}{=} \mathbf{1}\left[E(X_i, Y_i, U_i; \tilde{\pi}^{k(i),k(j)}) > E(X_j, Y_j, U_j; \tilde{\pi}^{k(i),k(j)})\right] \\
&= A_{ij}.
\end{aligned}
$$

Above, the second equality holds because the black-box estimators $\hat{\pi}^k$ are invariant to the ordering of their input data points, and the third equality in distribution holds because the data points $(X_i, Y_i, U_i)$ are exchangeable. Finally, we also trivially know that $A_{\sigma(i)\sigma(j)} = A_{ij}$ for any $i, j$ such that $k(i) = k(j)$.

$\qquad\square$

**Lemma S1** (Proved in [2])**.** *Consider any partition of $\{1, \ldots, n+m\}$ points into $K+1$ folds $\mathcal{I}_1, \ldots, \mathcal{I}_{K+1}$, with $m = n/K$. If a random matrix $A \in \{0,1\}^{(n+m) \times (n+m)}$ satisfies $A \overset{d}{=} \Sigma A \Sigma^{\top}$ for any $(n+m) \times (n+m)$ permutation matrix $\Sigma$ that does not mix points assigned to different folds, then, for any fixed $\alpha \in (0,1)$,*

$$\mathbb{P}\left[(n+1) \in \mathcal{F}(A)\right] \leq 2\alpha + \min\left\{\frac{2(1-1/K)}{n/K+1}, \frac{1-K/n}{K+1}\right\}, \tag{S9}$$

*where the set $\mathcal{F}(A)$ is defined as in* (S8) *and depends on $\alpha$. In the special case where $K = n$, this bound simplifies to:*

$$\mathbb{P}\left[(n+1) \in \mathcal{F}(A)\right] \leq 2\alpha. \tag{S10}$$

*Proof of Theorem S1.* The proof is effectively identical to that of Theorem 3 in [2], by the same argument as in the proof of Theorem 2.

$\square$

*Proof of Theorem S2.* Fix any $y \in \mathcal{Y}$ and suppose $Y_{n+1} = y$. Since $(X_1, Y_1), \ldots, (X_{n+1}, Y_{n+1})$ are marginally exchangeable, it follows that $(X_{n+1}, Y_{n+1})$ is exchangeable with all data points in $\{i \in \mathcal{I}_2 : Y_i = y\}$. By construction of the prediction set in (S3), we know that

$$y \notin \hat{\mathcal{C}}_{n,\alpha}^{\mathrm{SC-lc}}(X_{n+1})$$

if and only if

$$E_{n+1} = \min\left\{\tau \in [0,1] : y \in \mathcal{S}(X_{n+1}, U_{n+1}; \hat{\pi}, \tau)\right\} > \hat{\tau} \geq \hat{Q}_{1-\alpha}^{(y)}(\{E_i\}_{i \in \mathcal{I}_2}).$$

Since all the conformity scores $E_{n+1}$ and $\{E_i\}_{i \in \mathcal{I}_2 : Y_i = y}$ are exchangeable, the probability of the above miscoverage event can be no larger than $\alpha$, by the same argument as in [3].

$\square$

## S3 Supplementary experiments with simulated data

### S3.1 Implementation details

We have applied the following black-box classification methods to estimate label probabilities:

- a support vector classifier (SVC) with linear kernel, as implemented by the `sklearn` Python package with default parameters;
- a random forest classifier (RFC) with 1000 estimators of maximum depth 5, as implemented by the `sklearn` Python package with default parameters (except for the maximum number of features considered at each split, which we set equal to $p$).

For the CQC method, we carry out quantile regression on the classification scores using the same deep neural network employed in [3].

For simplicity, we split the data into subsets of equal size for all methods, including ours whenever using split-conformal calibration (Section 2.2). No effort was made to optimize the size of the splits for any method, so the empirical comparisons are fair. We do not expect the results of our experiments to change meaningfully if the sizes of the sample splits are optimized, since our method has the advantage of requiring one fewer split, and it has a stronger optimality property in theory.

### S3.2 Experiments with multinomial model and inhomogeneous features

Figure S1: Different classification methods on simulated data with 10 classes, for different choices of calibration and black-box models. The results correspond to 100 independent experiments with 10000 training samples and 5000 test samples each. JK+ is omitted for computational reasons. Other details are as in Figure 1.

## S3.3 Experiments with heteroscedastic decision-tree model and discrete features

We set $p = 5$ and generate each sample of features $X \in \mathbb{R}^p$ independently as follows: $X_1 = +1$ w.p. $3/4$, and $X_1 = -1$ w.p. $1/4$; $X_2 = +1$ w.p. $3/4$, and $X_2 = -2$ w.p. $1/4$; $X_3 = +1$ w.p. $1/4$, and $X_3 = -2$ w.p. $1/2$; $X_4$ is uniformly distributed on $\{1, \ldots, 4\}$; and $X_5 \sim \mathcal{N}(0, 1)$. The labels $Y$ belong to one of 4 possible classes, and their conditional distribution given $X = x$ is given by the decision tree shown in Figure S2, which only depends on the first four features.

Figure S2: A toy model for $P_{Y|X}$ in a classification setting with 4 labels.

The performances of the different methods on data generated from this model are compared in Figure S3. Here, the size of the training sample is equal to 10000 and the size of the test sample is equal to 5000; all experiments are repeated 100 times. Since these training sets are fairly large, for computational convenience we do not apply the JK+ method; see Figure S4 for a comparison including JK+ with smaller sample sizes.

These results are qualitatively consistent with those from Section 3.2, confirming that our methods have good approximate conditional coverage compared to the alternatives while not suffering from a significant power loss. It is interesting to note that the conditional distribution of $Y \mid X$ is more complicated here than in the previous example, hence the reason for a larger sample size. Despite this large sample size, the SVC black-box is unable to learn good estimates of the class probabilities. This is why methods with marginal coverage have relatively low power and poor conditional coverage. By contrast, the RFC black-box can learn these class probabilities quite accurately, and thus it allows our SC and CV+ methods to perform on par with the oracle (especially CV+, as expected). Again, the alternative methods do not achieve conditional coverage even with the help of the oracle.

Figure S3: Performance of alternative classification methods on simulated data with 4 classes. Results from 100 independent experiments with 10000 training samples and 5000 test samples each. Other details as in Figure 1.

Figure S4: Performance of alternative classification methods on simulated data with 4 classes. Results from 100 independent experiments with 1000 training samples and 5000 test samples each. Other details are as in Figure S3.

## S4 Supplementary experiments with real data

We compare the performance of our methods to that of HCC and CQC on four popular benchmark data sets:

1. MNIST is a handwritten digit classification data set, containing 60000 grayscale images of size $28 \times 28$ pixels, each associated with one of $C = 10$ classes. As a pre-processing step, we apply Principal Component Analysis (PCA) to each image, resulting in a feature vector $X$ of length $p = 50$.

2. CIFAR10 is another image classification data set. The data includes 50000 RGB images, each of size $32 \times 32 \times 3$, belonging to one of $C = 10$ classes. We also use PCA to reduce the dimension to $p = 50$.

3. Fashion-MNIST contains 60000 images associated with $C = 10$ classes of clothes. We run the same pre-processing step as in MNIST and CIFAR10, resulting in $p = 50$ features.

4. The task in the Mice Protein Expression data set[2] is to identify the class of a mouse based on genetic, behavioral and treatment covariates. After applying standard data cleaning, we have 1080 samples, $p = 77$ features, and $C = 8$ classes.

We use the same baseline predictive algorithms as in Section S3.1, although with slightly different RFC parameters—here the number of estimators is 100, and the minimum number of samples at a leaf node is 3. Additionally, we also consider a neural network (NNet) with one hidden layer of size 64 and ReLU activation function. We use the adam optimizer, with a minibatch of size 128, a learning rate of 0.01, and a total number of epochs equal to 20. The CQC method is implemented as described in Section S3.1. In the real data experiments we present a second variant of CQC, namely CQC-RF, where we replace the quantile neural network algorithm with quantile random forest. To this end, we use the default `skgarden` hyper-parameters for quantile random forest, except for the number of estimators and the minimum number of samples required to split an internal node, which we set to 100 and 3, respectively.

In the numerical experiments, we set the target coverage level to $90\%$ and compare the coverage, conditional coverage, and length of the different calibration methods combined with the above predictive algorithms. The performance metrics are averaged over 100 experiments. The results in Table S1 are obtained by randomly selecting $n_{\text{train}} \in \{500, 1000\}$ training examples from the Mice Protein Expression data set, used to fit and calibrate the predictive models. The remaining $n_{\text{test}} \in \{580, 80\}$ unseen samples formulate a test set, in which we evaluate the methods' performance. Tables S2, S3, and S4 correspond to MNIST, Fashion-MNIST, and CIFAR10 data sets. Each experiment is conducted by randomly selecting $n_{\text{train}} \in \{500, 1000, 5000, 10000\}$ training examples as well as a disjoint set of 5000 unseen test samples, selected at random.

In sum, all the calibration methods achieve an exact $90\%$ marginal coverage, as guaranteed by the theory. CV+ and JK+ tend to achieve conditional coverage as well (green colored numbers), and SC performs slightly worse. In contrast, in most cases CQC, CQC-RF, and HCC fail (red colored numbers) to obtain the desired conditional coverage. As for the statistical efficiently, HCC often results in the shortest prediction sets—while failing to attain conditional coverage. Here, our methods are typically competitive and can even produce smaller prediction sets in some cases.

## S5 Supplementary tables

Table S1: Results of experiments on the Mice Protein Expression data. Mean marginal coverage, conditional coverage (worst-slab), and average size of the prediction sets conditional on coverage. Experiments obtained with different training set sizes are shown in different groups of rows. Standard deviations are shown in round brackets. Red: conditional coverage values below 0.88; green: conditional coverage values above 0.9; blue: smallest mean prediction set conditional on coverage.

| Black box | Coverage | | | | | | Conditional coverage | | | | | | Size l coverage | | | | | |
|---|---|---|---|---|---|---|---|---|---|---|---|---|---|---|---|---|---|---|
| | SC | CV+ | JK+ | CQC | CQC-RF | HCC | SC | CV+ | JK+ | CQC | CQC-RF | HCC | SC | CV+ | JK+ | CQC | CQC-RF | HCC |
| **500** | | | | | | | | | | | | | | | | | | |
| NNet | 0.90 (0.02) | 0.90 (0.02) | 0.90 (0.02) | 0.90 (0.02) | 0.91 (0.03) | 0.91 (0.02) | 0.89 (0.06) | 0.91 (0.05) | 0.91 (0.06) | 0.79 (0.11) | 0.81 (0.10) | 0.83 (0.09) | 1.11 (0.03) | 1.04 (0.01) | 1.03 (0.01) | 1.96 (1.23) | 1.08 (0.10) | 1.00 (0.00) |
| RFC | 0.90 (0.02) | 0.92 (0.02) | 0.91 (0.02) | 0.90 (0.02) | 0.91 (0.02) | 0.90 (0.02) | 0.86 (0.07) | 0.91 (0.05) | 0.89 (0.06) | 0.78 (0.12) | 0.83 (0.09) | 0.80 (0.10) | 1.54 (0.12) | 1.36 (0.05) | 1.30 (0.05) | 3.43 (1.33) | 1.95 (0.33) | 1.22 (0.09) |
| SVC | 0.90 (0.02) | 0.90 (0.02) | 0.90 (0.02) | 0.90 (0.03) | 0.90 (0.03) | 0.90 (0.02) | 0.89 (0.06) | 0.90 (0.06) | 0.90 (0.05) | 0.78 (0.12) | 0.80 (0.11) | 0.80 (0.10) | 1.22 (0.06) | 1.08 (0.02) | 1.07 (0.02) | 1.93 (1.24) | 1.05 (0.07) | 1.00 (0.00) |
| **1000** | | | | | | | | | | | | | | | | | | |
| NNet | 0.89 (0.02) | 0.91 (0.02) | 0.91 (0.04) | 0.90 (0.04) | 0.90 (0.03) | 0.90 (0.02) | NA (NA) | 0.91 (0.13) | NA (NA) | 0.91 (0.11) | NA (NA) | NA (NA) | 1.03 (0.02) | 1.01 (0.01) | 1.01 (0.01) | 1.00 (0.00) | 1.00 (0.00) | 1.00 (0.00) |
| RFC | 0.90 (0.03) | 0.90 (0.03) | 0.89 (0.03) | 0.89 (0.04) | 0.89 (0.04) | 0.90 (0.03) | NA (NA) | NA (NA) | NA (NA) | 0.92 (0.13) | NA (NA) | NA (NA) | 1.31 (0.06) | 1.19 (0.04) | 1.17 (0.06) | 1.92 (0.79) | 1.19 (0.10) | 1.01 (0.01) |
| SVC | 0.89 (0.04) | 0.90 (0.03) | 0.90 (0.03) | 0.90 (0.03) | 0.91 (0.04) | 0.90 (0.03) | NA (NA) | NA (NA) | NA (NA) | NA (NA) | NA (NA) | NA (NA) | 1.06 (0.04) | 1.02 (0.02) | 1.02 (0.02) | 1.10 (0.65) | 1.00 (0.00) | 1.00 (0.00) |

Table S2: Results of experiments on MNIST data. Other details are as in Table S1.

| Black box | Coverage | | | | | | Conditional coverage | | | | | | Size \| coverage | | | | | |
|---|---|---|---|---|---|---|---|---|---|---|---|---|---|---|---|---|---|---|
| | SC | CV+ | JK+ | CQC | CQC-RF | HCC | SC | CV+ | JK+ | CQC | CQC-RF | HCC | SC | CV+ | JK+ | CQC | CQC-RF | HCC |
| **500** | | | | | | | | | | | | | | | | | | |
| NNet | 0.90 (0.02) | 0.92 (0.01) | 0.90 (0.01) | 0.90 (0.02) | 0.90 (0.02) | 0.90 (0.02) | 0.85 (0.04) | 0.90 (0.03) | 0.88 (0.03) | 0.81 (0.05) | 0.86 (0.04) | 0.86 (0.04) | 2.51 (0.68) | 1.67 (0.08) | 1.54 (0.07) | 6.63 (1.29) | 3.17 (0.61) | 2.25 (0.39) |
| RFC | 0.91 (0.02) | 0.93 (0.01) | 0.92 (0.01) | 0.90 (0.02) | 0.90 (0.02) | 0.90 (0.02) | 0.88 (0.03) | 0.88 (0.03) | 0.88 (0.03) | 0.80 (0.07) | 0.85 (0.04) | 0.86 (0.03) | 2.55 (0.33) | 2.21 (0.13) | 2.15 (0.13) | 7.14 (1.77) | 4.02 (0.89) | 2.06 (0.31) |
| SVC | 0.90 (0.02) | 0.92 (0.01) | 0.90 (0.01) | 0.90 (0.02) | 0.90 (0.02) | 0.90 (0.02) | 0.87 (0.04) | 0.91 (0.02) | 0.89 (0.02) | 0.79 (0.05) | 0.83 (0.05) | 0.85 (0.04) | 2.51 (0.29) | 2.19 (0.12) | 2.06 (0.13) | 6.68 (1.52) | 3.97 (0.89) | 1.91 (0.31) |
| **1000** | | | | | | | | | | | | | | | | | | |
| NNet | 0.90 (0.01) | 0.93 (0.01) | 0.91 (0.01) | 0.90 (0.01) | 0.90 (0.01) | 0.90 (0.01) | 0.88 (0.03) | 0.92 (0.02) | 0.90 (0.02) | 0.83 (0.05) | 0.86 (0.04) | 0.86 (0.03) | 1.53 (0.09) | 1.43 (0.04) | 1.29 (0.03) | 5.37 (1.47) | 1.74 (0.24) | 1.36 (0.11) |
| RFC | 0.90 (0.01) | 0.92 (0.01) | 0.92 (0.01) | 0.90 (0.02) | 0.90 (0.01) | 0.90 (0.01) | 0.87 (0.03) | 0.88 (0.03) | 0.88 (0.03) | 0.81 (0.05) | 0.87 (0.03) | 0.86 (0.04) | 2.07 (0.13) | 1.99 (0.08) | 1.96 (0.08) | 6.17 (1.59) | 2.47 (0.44) | 1.53 (0.13) |
| SVC | 0.90 (0.01) | 0.92 (0.01) | 0.90 (0.01) | 0.90 (0.01) | 0.90 (0.01) | 0.90 (0.01) | 0.89 (0.03) | 0.92 (0.02) | 0.90 (0.02) | 0.81 (0.04) | 0.86 (0.04) | 0.86 (0.03) | 2.07 (0.11) | 1.91 (0.08) | 1.82 (0.08) | 5.63 (1.58) | 1.89 (0.36) | 1.37 (0.11) |
| **5000** | | | | | | | | | | | | | | | | | | |
| NNet | 0.90 (0.01) | 0.92 (0.01) | NA | 0.90 (0.01) | 0.90 (0.01) | 0.90 (0.01) | 0.90 (0.02) | 0.92 (0.01) | NA | 0.87 (0.03) | 0.87 (0.03) | 0.88 (0.03) | 1.13 (0.01) | 1.14 (0.01) | NA | 1.14 (0.46) | 1.02 (0.02) | 1.00 (0.00) |
| RFC | 0.90 (0.01) | 0.91 (0.00) | NA | 0.90 (0.01) | 0.90 (0.01) | 0.90 (0.01) | 0.87 (0.03) | 0.89 (0.02) | NA | 0.85 (0.04) | 0.88 (0.02) | 0.87 (0.03) | 1.80 (0.05) | 1.76 (0.03) | NA | 3.05 (1.19) | 1.27 (0.06) | 1.11 (0.02) |
| SVC | 0.90 (0.01) | 0.91 (0.01) | NA | 0.90 (0.01) | 0.90 (0.01) | 0.90 (0.01) | 0.90 (0.02) | 0.91 (0.02) | NA | 0.85 (0.04) | 0.87 (0.02) | 0.87 (0.03) | 1.61 (0.04) | 1.50 (0.03) | NA | 1.96 (1.28) | 1.09 (0.02) | 1.05 (0.01) |
| **10000** | | | | | | | | | | | | | | | | | | |
| NNet | 0.90 (0.01) | 0.92 (0.00) | NA | 0.90 (0.01) | 0.90 (0.01) | 0.90 (0.01) | 0.90 (0.02) | 0.92 (0.01) | NA | 0.87 (0.02) | 0.87 (0.02) | 0.87 (0.02) | 1.07 (0.01) | 1.10 (0.01) | NA | 1.00 (0.00) | 1.00 (0.00) | 1.00 (0.00) |
| RFC | 0.90 (0.01) | 0.91 (0.00) | NA | 0.90 (0.01) | 0.90 (0.01) | 0.90 (0.01) | 0.88 (0.02) | 0.89 (0.02) | NA | 0.87 (0.03) | 0.88 (0.02) | 0.87 (0.03) | 1.74 (0.03) | 1.70 (0.02) | NA | 1.59 (0.60) | 1.13 (0.02) | 1.05 (0.01) |
| SVC | 0.90 (0.01) | 0.91 (0.01) | NA | 0.90 (0.01) | 0.90 (0.01) | 0.90 (0.01) | 0.90 (0.02) | 0.91 (0.02) | NA | 0.87 (0.03) | 0.87 (0.03) | 0.87 (0.03) | 1.46 (0.02) | 1.38 (0.02) | NA | 1.07 (0.07) | 1.04 (0.01) | 1.02 (0.00) |

Table S3: Results of experiments on Fashion-MNIST data. Other details are as in Table S1.

| Black box | Coverage | | | | | | Conditional coverage | | | | | | Size \| coverage | | | | | |
|---|---|---|---|---|---|---|---|---|---|---|---|---|---|---|---|---|---|---|
| | SC | CV+ | JK+ | CQC | CQC-RF | HCC | SC | CV+ | JK+ | CQC | CQC-RF | HCC | SC | CV+ | JK+ | CQC | CQC-RF | HCC |
| **500** | | | | | | | | | | | | | | | | | | |
| NNet | 0.90 (0.02) | 0.92 (0.01) | 0.90 (0.01) | 0.91 (0.03) | 0.91 (0.02) | 0.90 (0.02) | 0.86 (0.04) | 0.88 (0.03) | 0.86 (0.03) | 0.79 (0.06) | 0.85 (0.04) | 0.85 (0.04) | 2.97 (0.78) | 1.96 (0.22) | 1.82 (0.17) | 6.59 (1.24) | 3.59 (0.69) | 2.60 (0.55) |
| RFC | 0.90 (0.02) | 0.92 (0.01) | 0.92 (0.01) | 0.90 (0.03) | 0.90 (0.02) | 0.91 (0.02) | 0.88 (0.03) | 0.91 (0.02) | 0.90 (0.02) | 0.76 (0.07) | 0.82 (0.06) | 0.86 (0.05) | 2.47 (0.24) | 2.28 (0.11) | 2.23 (0.10) | 6.68 (1.77) | 3.95 (0.83) | 1.98 (0.22) |
| SVC | 0.90 (0.02) | 0.93 (0.01) | 0.90 (0.02) | 0.90 (0.03) | 0.90 (0.02) | 0.90 (0.02) | 0.86 (0.03) | 0.91 (0.02) | 0.88 (0.03) | 0.76 (0.06) | 0.80 (0.05) | 0.84 (0.05) | 2.74 (0.29) | 2.43 (0.13) | 2.30 (0.14) | 6.61 (1.41) | 4.87 (0.80) | 2.15 (0.29) |
| **1000** | | | | | | | | | | | | | | | | | | |
| NNet | 0.90 (0.01) | 0.93 (0.01) | 0.90 (0.01) | 0.90 (0.02) | 0.90 (0.02) | 0.90 (0.01) | 0.85 (0.04) | 0.91 (0.02) | 0.87 (0.03) | 0.81 (0.06) | 0.85 (0.04) | 0.85 (0.04) | 1.82 (0.21) | 1.71 (0.06) | 1.53 (0.05) | 5.73 (1.57) | 2.13 (0.31) | 1.72 (0.15) |
| RFC | 0.90 (0.01) | 0.92 (0.01) | 0.91 (0.01) | 0.90 (0.01) | 0.90 (0.01) | 0.90 (0.01) | 0.89 (0.03) | 0.91 (0.02) | 0.90 (0.02) | 0.77 (0.06) | 0.84 (0.04) | 0.85 (0.04) | 2.20 (0.13) | 2.11 (0.07) | 2.09 (0.07) | 5.88 (1.73) | 2.70 (0.55) | 1.64 (0.10) |
| SVC | 0.90 (0.02) | 0.92 (0.01) | 0.90 (0.01) | 0.90 (0.02) | 0.90 (0.02) | 0.90 (0.01) | 0.88 (0.02) | 0.91 (0.02) | 0.88 (0.02) | 0.78 (0.05) | 0.83 (0.04) | 0.85 (0.04) | 2.30 (0.13) | 2.11 (0.07) | 2.01 (0.08) | 5.89 (1.60) | 3.26 (0.54) | 1.71 (0.12) |
| **5000** | | | | | | | | | | | | | | | | | | |
| NNet | 0.90 (0.01) | 0.94 (0.01) | NA | 0.90 (0.01) | 0.90 (0.01) | 0.90 (0.01) | 0.86 (0.03) | 0.93 (0.02) | NA | 0.84 (0.04) | 0.86 (0.03) | 0.86 (0.03) | 1.35 (0.02) | 1.45 (0.02) | NA | 3.29 (1.40) | 1.36 (0.05) | 1.29 (0.04) |
| RFC | 0.90 (0.01) | 0.91 (0.01) | NA | 0.90 (0.01) | 0.90 (0.01) | 0.90 (0.01) | 0.89 (0.02) | 0.91 (0.01) | NA | 0.81 (0.06) | 0.87 (0.03) | 0.86 (0.03) | 1.92 (0.04) | 1.87 (0.02) | NA | 3.91 (1.50) | 1.65 (0.10) | 1.32 (0.03) |
| SVC | 0.90 (0.01) | 0.91 (0.01) | NA | 0.90 (0.01) | 0.90 (0.01) | 0.90 (0.01) | 0.89 (0.02) | 0.91 (0.02) | NA | 0.82 (0.04) | 0.85 (0.03) | 0.85 (0.03) | 1.76 (0.04) | 1.67 (0.03) | NA | 3.49 (1.23) | 1.74 (0.16) | 1.29 (0.03) |
| **10000** | | | | | | | | | | | | | | | | | | |
| NNet | 0.90 (0.01) | 0.94 (0.00) | NA | 0.90 (0.01) | 0.90 (0.01) | 0.90 (0.01) | 0.88 (0.03) | 0.94 (0.02) | NA | 0.87 (0.03) | 0.86 (0.03) | 0.86 (0.03) | 1.31 (0.02) | 1.43 (0.02) | NA | 1.47 (0.38) | 1.26 (0.03) | 1.22 (0.02) |
| RFC | 0.90 (0.01) | 0.91 (0.01) | NA | 0.90 (0.01) | 0.90 (0.01) | 0.90 (0.01) | 0.89 (0.02) | 0.91 (0.02) | NA | 0.86 (0.04) | 0.88 (0.03) | 0.87 (0.03) | 1.83 (0.03) | 1.79 (0.02) | NA | 2.20 (0.90) | 1.49 (0.06) | 1.25 (0.02) |
| SVC | 0.90 (0.01) | 0.91 (0.01) | NA | 0.90 (0.01) | 0.90 (0.01) | 0.90 (0.01) | 0.89 (0.02) | 0.90 (0.02) | NA | 0.85 (0.03) | 0.86 (0.03) | 0.86 (0.03) | 1.63 (0.03) | 1.57 (0.02) | NA | 1.85 (0.72) | 1.40 (0.06) | 1.21 (0.02) |

Table S4: Results of experiments on CIFAR10 data. Other details are as in Table S1.

| Black box | Coverage | | | | | | Conditional coverage | | | | | | Size \| coverage | | | | | |
|---|---|---|---|---|---|---|---|---|---|---|---|---|---|---|---|---|---|---|
| | SC | CV+ | JK+ | CQC | CQC-RF | HCC | SC | CV+ | JK+ | CQC | CQC-RF | HCC | SC | CV+ | JK+ | CQC | CQC-RF | HCC |
| **500** | | | | | | | | | | | | | | | | | | |
| NNet | 0.91 (0.02) | 0.94 (0.01) | 0.92 (0.01) | 0.90 (0.02) | 0.90 (0.03) | 0.90 (0.02) | 0.88 (0.03) | 0.93 (0.02) | 0.90 (0.02) | 0.77 (0.06) | 0.80 (0.06) | 0.85 (0.04) | 8.21 (0.37) | 8.43 (0.23) | 7.99 (0.25) | 9.55 (0.16) | 9.37 (0.19) | 9.05 (0.28) |
| RFC | 0.90 (0.02) | 0.96 (0.01) | 0.96 (0.01) | 0.91 (0.02) | 0.90 (0.03) | 0.90 (0.02) | 0.90 (0.02) | 0.96 (0.01) | 0.96 (0.02) | 0.79 (0.06) | 0.84 (0.05) | 0.90 (0.03) | 7.80 (0.36) | 8.23 (0.32) | 8.11 (0.34) | 9.43 (0.44) | 8.67 (0.40) | 7.67 (0.37) |
| SVC | 0.90 (0.02) | 0.93 (0.01) | 0.91 (0.02) | 0.90 (0.03) | 0.90 (0.02) | 0.90 (0.02) | 0.89 (0.03) | 0.92 (0.02) | 0.89 (0.03) | 0.80 (0.06) | 0.82 (0.05) | 0.87 (0.03) | 7.92 (0.37) | 7.88 (0.32) | 7.41 (0.38) | 9.39 (0.47) | 8.75 (0.41) | 7.83 (0.33) |
| **1000** | | | | | | | | | | | | | | | | | | |
| NNet | 0.90 (0.01) | 0.96 (0.01) | 0.93 (0.01) | 0.90 (0.02) | 0.90 (0.02) | 0.90 (0.01) | 0.88 (0.03) | 0.95 (0.01) | 0.92 (0.02) | 0.77 (0.04) | 0.82 (0.04) | 0.86 (0.03) | 7.78 (0.25) | 8.08 (0.18) | 7.82 (0.17) | 9.47 (0.16) | 9.19 (0.15) | 8.80 (0.23) |
| RFC | 0.90 (0.01) | 0.96 (0.01) | 0.96 (0.01) | 0.90 (0.02) | 0.90 (0.02) | 0.90 (0.01) | 0.90 (0.02) | 0.96 (0.01) | 0.96 (0.01) | 0.80 (0.06) | 0.84 (0.04) | 0.90 (0.02) | 7.11 (0.31) | 7.55 (0.24) | 7.47 (0.24) | 9.43 (0.45) | 8.16 (0.42) | 6.90 (0.30) |
| SVC | 0.90 (0.01) | 0.92 (0.01) | 0.90 (0.01) | 0.90 (0.02) | 0.90 (0.02) | 0.90 (0.01) | 0.88 (0.02) | 0.92 (0.02) | 0.89 (0.02) | 0.80 (0.05) | 0.82 (0.04) | 0.88 (0.03) | 7.35 (0.27) | 7.18 (0.23) | 6.75 (0.24) | 9.40 (0.42) | 8.58 (0.43) | 7.22 (0.31) |
| **5000** | | | | | | | | | | | | | | | | | | |
| NNet | 0.90 (0.01) | 0.96 (0.00) | NA | 0.90 (0.01) | 0.90 (0.01) | 0.90 (0.01) | 0.89 (0.02) | 0.96 (0.01) | NA | 0.79 (0.05) | 0.84 (0.03) | 0.86 (0.02) | 7.09 (0.14) | 7.25 (0.12) | NA | 9.16 (0.20) | 8.40 (0.14) | 7.86 (0.19) |
| RFC | 0.90 (0.01) | 0.95 (0.00) | NA | 0.90 (0.01) | 0.90 (0.01) | 0.90 (0.01) | 0.89 (0.02) | 0.94 (0.01) | NA | 0.82 (0.04) | 0.86 (0.03) | 0.90 (0.02) | 6.03 (0.15) | 6.36 (0.11) | NA | 8.81 (0.51) | 7.33 (0.29) | 5.81 (0.15) |
| SVC | 0.90 (0.01) | 0.91 (0.01) | NA | 0.90 (0.01) | 0.90 (0.01) | 0.90 (0.01) | 0.89 (0.02) | 0.90 (0.02) | NA | 0.81 (0.04) | 0.84 (0.03) | 0.89 (0.02) | 6.09 (0.12) | 5.99 (0.10) | NA | 8.84 (0.41) | 7.75 (0.29) | 5.96 (0.12) |
| **10000** | | | | | | | | | | | | | | | | | | |
| NNet | 0.90 (0.01) | 0.96 (0.00) | NA | 0.90 (0.01) | 0.90 (0.01) | 0.90 (0.01) | 0.89 (0.02) | 0.95 (0.01) | NA | 0.81 (0.05) | 0.85 (0.03) | 0.87 (0.02) | 5.99 (0.16) | 5.89 (0.09) | NA | 8.89 (0.26) | 7.92 (0.12) | 5.93 (0.24) |
| RFC | 0.90 (0.01) | 0.94 (0.00) | NA | 0.90 (0.01) | 0.90 (0.01) | 0.90 (0.01) | 0.90 (0.02) | 0.94 (0.01) | NA | 0.83 (0.04) | 0.86 (0.03) | 0.89 (0.02) | 5.69 (0.09) | 5.99 (0.05) | NA | 8.18 (0.63) | 7.13 (0.22) | 5.44 (0.09) |
| SVC | 0.90 (0.01) | 0.91 (0.00) | NA | 0.90 (0.01) | 0.90 (0.01) | 0.90 (0.01) | 0.89 (0.02) | 0.89 (0.02) | NA | 0.82 (0.03) | 0.84 (0.03) | 0.89 (0.02) | 5.87 (0.08) | 5.83 (0.06) | NA | 8.37 (0.48) | 7.66 (0.18) | 5.74 (0.08) |

## Footnotes

[2]`https://archive.ics.uci.edu/ml/datasets/Mice+Protein+Expression`