[Reviews · NeurIPS 2020]

Review 1

Summary and Contributions: The paper proposes a new conformity score for categorical / unordered response labels that is adaptive to the heterogeneity of difficulty of examples. The key insight of using generalized inverse quantile conformity measure E has the potential to be useful in settings beyond the classification problem discussed in the paper. UPDATE--------------------------------------------------------------------------------------- I thank the authors for their feedback. Both my questions were addressed quite satisfactorily.

Strengths: The novel conformity score Eq. (7) (via Eq. (3)) is an elegant solution to the problem of heterogeneity in classification setting. It has several advantages compared to the existing approaches, such as HCC (which may have quite an uneven coverage across the feature space) or CQC (which requires a three-way split and a quantile regression method on top of a score fitting method). The authors also make a good point that the key idea is applicable beyond the classification setting. For example, it is now easier to extend conformal prediction framework to the set-ups where Y may be multidimensional.

Weaknesses: At the time of this writing, I cannot think of any significant weaknesses that are inherent to the new conformity score.

Correctness: Yes, I took a brief pass at the proofs in the supplement, and they appear to contain no errors. (As the authors remark, the proofs themselves are quite standard in this literature.)

Clarity: The paper is well written and quite easy to follow. Prefacing the introduction of the new conformity score with the discussion of the oracle method in Section 1.1 is a nice touch.

Relation to Prior Work: The prior works are discussed adequately.

Reproducibility: Yes

Additional Feedback: Questions: - Could more details be given about the exact implementation of the competing methods, HCC, CQC and CQC-RF for the experiments in Sections 3 and 4? For example, CQC (and CQC-RF) requires a three-way split. Were any efforts made to tune the size of the splits? - Another coverage guarantee that may be of interest in the classification setting is label-conditional coverage guarantee (as in [24] Vovk et al. (2005), [19] Sadinle et al. (2019)). Can this be achieved with the proposed score?


Review 2

Summary and Contributions: This paper develops specialized versions of some recent developments in predictive inference, eg [1,4,18], for categorical and unordered response. More precisely, the author(s) focus on valid classification coverage using black-box classifier (assumed to treat all samples exchangeably). To this end, they use techniques from conformal prediction and introduce a new conformity score. Their (randomized) prediction sets are built mimicking the oracle prediction set using this new conformity score (7). This new conformity score is the cornerstone of their methodology and an interesting contribution. Although it appears to be a simple modification from hindsight, it is non-trivial from foresight because their "generalized quantile" based method is highly adaptive to the distributional heterogeneity, which is ubiquitous in real-world applications. Using ground-breaking results of [4,18] and a reduction argument (S6) in the supplement, the author(s) prove marginal coverage for split-conformal calibration, CV+ calibration and jackknife+ calibration. The papers presents fine experiments assessing the advantages of the proposed methods over existing alternatives.

Strengths: On the theoretical side, proving valid coverage is a difficult task especially when using CV+ calibration. Recent advances [4,18] might furnish some important tools to overcome this difficulty. This paper manages to reduce the classification problem to the above framework of the recent advances. This simple but not elementary step is the missing key to unveil powerful theoretical results (Theorem 1 and 2). On the empirical evaluation side, the paper gives a quite comprehensive study on the benefits of their methods.

Weaknesses: One potential weakness might be the positioning of this paper in regards of existing works. This paper might be seen as a specialized version of the papers [1,4,18] since most of the theoretical hardness and ideas are dealt by these references. But one might also argue that the conformity score and the reduction step (S6) are important key notion/step brought by this submission. Furthermore, their numerical experiments show that their proposed methods enjoy very nice statistical properties. (see also "relation to prior work" below)

Correctness: Claims and method are correct. On the methodology side, the author(s) use the notion of Worst (Conditional) Slice to evaluate the conditional coverage. This method is fine and well presented in the supplement.

Clarity: The paper is well written.

Relation to Prior Work: A clear discussion on previous works is given in the paper. In particular, a recent paper [2] offers similar guarantees applying quantile regression of hold-out scores of black-box classifiers. The experiments of the present paper tends to suggest that the author(s)'s methods might be superior. More fundamentally, the author(s) argue that their method is more adaptive to the distributional heterogeneity (end of Section 2.4), and I agree. [I agree with the authors' answer on this point]

Reproducibility: Yes

Additional Feedback:


Review 3

Summary and Contributions: * a new technique for obtaining calibrated probabilistic predictions for classification * theoretical justification, showing the method is near-optimal * experimental validation

Strengths: + neat idea! (use the smallest set of labels that contains exactly the required probability mass using a simple randomization trick) + theoretical guarantee of near optimality + excellent empirical performance + relatively well-written

Weaknesses: – will still require some more explaining to be accessible by the wider NeurIPS community – experiments are based on relatively simple and "boring" benchmark data sets (MNIST; CIFAR, ...), but on the other hand, at least they are nice and clean

Correctness: as far as I can tell

Clarity: mostly (but as I said, explaining the intuitions behind conformal prediction in more detail will help)

Relation to Prior Work: yes. I don't see a problem with the parallel work [2] not being carefully discussed since the [2] is so recent (1st version April 21, 2020 arXiv) that the authors haven't had a chance to study it properly

Reproducibility: Yes

Additional Feedback: ***I have read the author rebuttal. I have no further remarks based on it. I will keep my score unchanged.*** - is it really valid to say that the independence assumption (in i.i.d.) is unnecessary? that would sound like any identically distributed data are exchangeable, which isn't the case - the idea in the generalized inverse S is quite intuitive and can be understood with a few readings of the paragraph following its definition, but it's very hard to decipher while reading the definition itself. it'd be helpful to give the intuition first so that the reader can better understand the definition. - p. 2, l. 49 (about discarding zero probability labels): I don't see how zero probability elements can enter the set C^oracle in the first place - similarly to S, the intuitive definition of E, Eq. (7), would be helpful along the lines of "the smallest probability \tau s.t. $y$ is a member of the generalized inverse S" - C^sc, Eq. (8), is defined inside Algorithm 1. it took me a while to find it there. will be helpful to point this out to the reader in the text

[Author Response · NeurIPS 2020]

We thank the reviewers for taking the time to read our manuscript carefully, as well as for providing very insightful and constructive comments. We are flattered by their mostly positive feedback (e.g., "neat idea!", "excellent empirical performance", "elegant solution", "potential to be useful", "mostly well written") and we welcome their suggestions on how to improve the writing to make the paper more accessible. We respond to the reviewers' comments in detail below. We feel that the suggested improvements are relatively few and will be easy to implement in a minor revision.

## Extensions

*Label-conditional coverage.* To answer the reviewer's interesting comment: yes, provable label-conditional coverage can be easily achieved by calibrating the threshold $\tau$ separately for each class. More precisely, focusing on the extension of Algorithm 1 for simplicity, we would compute $\hat{Q}_{1-\alpha}^{(y)}(\{E_i\}_{i\in\mathcal{I}_2})$ as the $\lceil(1-\alpha)(1+|\{i \in \mathcal{I}_2 : Y_i = y\}|)\rceil$th largest value in $\{E_i\}_{i\in\mathcal{I}_2:Y_i=y}$, for each $y \in \mathcal{Y}$. Then, we would define $\hat{\tau} = \max_{y\in\mathcal{Y}} \hat{Q}_{1-\alpha}^{(y)}(\{E_i\}_{i\in\mathcal{I}_2})$ and output $\hat{\mathcal{C}}_{n,\alpha}^{\mathrm{sc}}(X_{n+1}) = \mathcal{S}(X_{n+1}, U_{n+1}; \hat{\pi}, \hat{\tau})$. We would be happy to include this extension in a revised manuscript.

## Minor comments

*Size of data splits for CQC.* We agree with the reviewer that this point should be clarified. For simplicity, we split the data into subsets of equal size for all methods, including ours (Section 2.2). No effort was made to optimize the size of the splits for any method, so the empirical comparisons are fair. We do not expect the results of our experiments to change meaningfully if the sizes of the sample splits are optimized, since our method has the advantage of requiring one fewer split, and it has a stronger optimality property in theory.

*Relation with prior work.* As highlighted by the reviewer, it is clear that the prediction sets in (5) and the conformity scores in (7) are the novel contributions of our paper. Our method indeed builds upon the theory on model-free predictive inference previously developed by others, and we gratefully acknowledge those works. That said, to make conformal inference useful, one needs high-quality conformity scores leading to tight prediction sets, which is the focus of this paper and the current research frontier.

*Choice of data sets.* The reviewer correctly points out that our data sets are "standard" and in that sense "quite boring". However, we feel that our choice is well-justified because our goal is to make the results easily accessible to the largest possible audience without distracting from the methodological message of the paper. Furthermore, choosing an unusual data set for comparing methods, without a good reason, may give the wrong impression that the example is somehow "cherry-picked".

*Independence vs. exchangeability.* Our wording when we said "independence is unnecessary" was accidentally a little ambiguous. What we meant is that "exchangeability" is sufficient. Our assumptions are stated explicitly later in the paper, but we acknowledge that the sentence "independence is unnecessary" at the beginning should be clarified. We thank the reviewer for bringing this issue to our attention.

*Labels with zero probability.* The referee is right: our comment about zero-probability elements can be removed. We thank the reviewer for suggesting this improvement.

## Exposition

*Definition of the generalized inverse $\mathcal{S}$.* We would be happy to provide some intuition for this definition before the formal statement, following the referee's suggestion.

*Definition of $E$ in (7).* We would gladly follow the referee's suggestion to describe $E$ in words with something along the lines of "the smallest probability $\tau$ s.t. $y$ is a member of the generalized inverse $\mathcal{S}$".

*Referencing (8) in the text.* We agree that (8) should be referenced more clearly in the text; we will gladly do so in the revised manuscript.

[Meta-Review · NeurIPS 2020]

All reviewers found the novel conformity score to be an "elegant solution to the problem of heterogeneity in classification setting." Proving valid coverage is difficult in this setting, and the paper leverages recent advances to do exactly that. As R4 states, I agree that the work could do with paper revisions that would open it up to be more accessible to the wider NeurIPS community. At the moment, the work is fairly niche in its own community, e.g., it uses non-ML empirical standards of synthetic to MNIST scale.